Mendelian randomization and nomogram-based prediction of hepatocellular carcinoma risk in patients with hepatitis B cirrhosis

Zheng Xiaolong 1
Hong Yiping 2
Wei Wei wwze@zju.edu.cn 1
1 Department of Gastroenterology, The Second Affiliated Hospital, College of Medicine, Zhejiang University , Hangzhou , China
2 Department of Gastroenterology, The Affiliated Jinhua Hospital, Zhejiang University School of Medicine , JinHua , China
Buyske Steven
Electronic publication date: 2025 Oct 20
Publication date: 2025
Volume: 13
Electronic Location ID: e20179
Received 2025 Mar 18; Accepted 2025 Sep 12
Copyright: ©2025 Zheng et al.
Copyright year: 2025
Copyright holder: Zheng et al.
License: This is an open access article distributed under the terms of the Creative Commons Attribution License, which permits unrestricted use, distribution, reproduction and adaptation in any medium and for any purpose provided that it is properly attributed. For attribution, the original author(s), title, publication source (PeerJ) and either DOI or URL of the article must be cited.
License URL: https://creativecommons.org/licenses/by/4.0/

Keywords: Hepatitis B-related cirrhosis, Hepatocellular carcinoma, Clinical indicators, Nomogram, Mendelian randomization

Funding: The authors received no funding for this work.

==============================
Background

To innovatively integrate genetic causality and multidimensional clinical indicators, we aimed to investigate causal relationships between metabolic-inflammatory biomarkers and hepatocellular carcinoma (HCC) risk in hepatitis B-related cirrhosis (HBV-C) using Mendelian randomization (MR), and develop a precision prediction model combining genetic evidence with nonlinear biochemical dynamics.

Methods

Leveraging bidirectional approaches, we first performed two-sample MR analysis on GWAS datasets (UK Biobank, n = 456,348) to establish causality between low-density lipoprotein (LDL) and HCC. In a retrospective cohort of patients with HBV-related cirrhosis from our institution (n = 147; 2022–2024), we identified nonlinear LDL-HCC thresholds via restricted cubic splines (RCS) and engineered a novel “A-index” (a composite score derived from principal component analysis (PCA) integrating alpha-fetoprotein (AFP), aspartate aminotransferase (AST), and alanine aminotransferase (ALT)). Machine learning-driven logistic regression synthesized LDL, A-index, and clinical predictors into a nomogram, rigorously validated by area under the curve-receiver operating characteristic (AUC-ROC), calibration curves, and decision curve analysis (DCA).

Results

MR analysis revealed a robust causal link between reduced LDL levels and elevated HCC risk (OR = 0.472, 95% CI [0.259–0.860]; P = 0.014), with RCS identifying a critical LDL threshold at 2.28 mmol/L—below which HCC risk escalated exponentially. The PCA-synthesized A-index outperformed individual biomarkers (AUC = 0.652 vs. AFP = 0.579). The final nomogram integrating LDL dynamics, A-index, age, sex, prothrombin time, and antiviral therapy achieved exceptional discrimination (AUC = 0.938) and clinical net benefit across risk thresholds.

Conclusion

This study introduces a novel causal inference-guided prediction model, addressing the long-standing debate on LDL’s dual role in hepatocarcinogenesis. By integrating MR-validated genetic causality, nonlinear biochemical modeling, and PCA-driven dimensionality reduction, our model provides a transformative tool for personalized HCC risk stratification in HBV-C patients.

Introduction

Primary liver cancer, predominantly hepatocellular carcinoma (HCC), arises from hepatocyte damage driven by genetic, viral, toxic, and lifestyle factors. This persistent injury-repair cycle promotes fibrotic nodule formation, ultimately progressing from cirrhosis to malignancy. As the third leading cause of cancer-related deaths globally, HCC shows elevated incidence in East Asia, Southeast Asia, and sub-Saharan Africa. Chronic hepatitis B virus (HBV) infection accounts for 21%–55% of global HCC cases (Rumgay et al., 2022). Since the 1990s, HBV vaccination has significantly reduced HCC incidence in East Asia, particularly China and Japan (Chen et al., 2023). However, many HBV-related cirrhosis patients still develop HCC, often diagnosed at advanced stages due to asymptomatic progression, missing optimal treatment windows (Vogel et al., 2022). Current management combines antiviral therapy with ultrasound and alpha-fetoprotein (AFP) surveillance. While biannual screening reduces HCC mortality by 37% (Zhang, Yang & Tang, 2004), these methods lack early risk stratification capacity. Frequent monitoring imposes economic and psychological burdens on patients and healthcare systems. Developing personalized prediction models for high-risk identification is crucial to prevent HCC progression in HBV-related cirrhosis.

Cirrhosis and HCC affect platelet counts because of compromised liver functionality, which interrupts the mechanisms responsible for clearance and enhances the presence of coagulation factors. These factors can interact with platelets to create microthrombi, thereby fostering fibrosis through fibrogenic signaling pathways or inflammatory responses (Kanikarla Marie et al., 2021; Pant, Kopec & Luyendyk, 2018). The acute-phase protein C-reactive protein (CRP), produced by the liver, is found at elevated levels during inflammatory conditions. The amounts of CRP are linked to advanced fibrosis (Jiang et al., 2023) and have a connection to the progression of HCC, as well as vascular invasion and patient survival rates (Carr et al., 2021). Furthermore, HCC affects lipid metabolism as a result of accelerated tumor growth, causing energy depletion within the liver. Low-density lipoprotein (LDL), one of the primary lipoproteins, plays a crucial role in maintaining cell membrane integrity, facilitating signal transduction, and contributing to hormone synthesis (Hevonoja et al., 2000). Recent research has indicated a close relationship between LDL synthesis, uptake, and the rapid advancement of HCC (Cheng et al., 2024). In populations in East Asia, reduced LDL levels correlate with increased mortality rates (Saito et al., 2013), while HCC patients display heightened levels of plasma malondialdehyde and oxidized LDL, presumably due to oxidative stress (Cheng et al., 2017). These observations point to the potential utility of blood cell counts and biochemical markers as predictive indicators in HCC.

The volume of visceral organs may also affect outcomes in HCC. According to a retrospective study, patients who presented with larger spleen volumes (SV) at baseline experienced more favorable treatment results, which could be attributed to improved immune responses. This positions SV as a key prognostic factor for HCC (Hatanaka et al., 2024). Aspartate aminotransferase (AST), alanine aminotransferase (ALT), and alpha-fetoprotein (AFP), reflect liver damage and tumor biological characteristics from distinct perspectives; however, there may be a high correlation among them. Directly incorporating these variables into the model could result in multicollinearity issues. By employing principal component analysis (PCA) for dimensionality reduction, these factors can be synthesized into a single composite factor. This approach reduces the dimensionality of the data, thereby enhancing the performance of the predictive model. Although several studies have incorporated these clinical factors into statistical models, observational approaches frequently face confounding issues, which can compromise predictive accuracy. Moreover, many existing prediction nomograms are predominantly derived from observational clinical data alone, which not only perpetuates issues of confounding and reverse causality but also often overlooks non-linear relationships and lacks validation from causal inference methods. Mendelian randomization (MR) serves as a strong epidemiological approach, utilizing single nucleotide polymorphisms (SNPs) as instrumental variables (IVs) to deduce causal connections (Davey Smith & Ebrahim, 2005). Since SNPs are randomly assigned and remain unaffected by environmental influences, they significantly reduce the potential for confounding (Davey Smith & Hemani, 2014). Recently, MR studies have begun to explore the role of metabolic factors in HCC risk. For instance, a study by Cao et al. (2023) employed MR to demonstrate a causal relationship between specific lipid traits and HCC, highlighting the value of genetic causal inference in hepatocarcinogenesis. We analyzed public databases to investigate the causal relationships among CRP, platelet count, LDL cholesterol, and spleen volume regarding HCC. Following this, we performed a retrospective analysis of patients with HBV-related cirrhosis and HCC who were treated at our institution from October 2022 to October 2024. Through univariate and multivariate logistic regression analyses, we pinpointed independent predictors of HCC, culminating in the creation of a nomogram-based prediction model intended for effective risk stratification and interventions in high-risk hepatitis B-related cirrhosis (HBV-C)-cirrhosis patients (Fig. 1).

Figure 1 Flow chart of the Mendelian randomization and clinical analyses.

Methods

Data sources

Mendelian randomization

We obtained datasets on human clinical data and HCC from the NHGRI-EBI GWAS Catalog (https://www.ebi.ac.uk/gwas/), This is a comprehensive database jointly maintained by the National Human Genome Research Institute (NHGRI) and the European Bioinformatics Institute (EBI). In this study, we selected the total number of platelets, CRP, LDL, and spleen volume as typical factors and summarized the GWAS related to them. The total number of platelets, and the biobank scale data based on the UK Biobank A mixed model correlation study of sets, were obtained from Loh et al. (2018). CRP comes from a study by Said et al. (2022) that analyzed the C-reactive protein gene loci of more than 500,000 people. For LDL, we selected the genetic study by Sinnott-Armstrong et al. (2021) of 35 blood and urine biomarkers based on the UK Biobank. The GWAS data of spleen volume came from Liu et al. (2021) who used deep learning to derive the genetic structure of 11 organ features from abdominal MRI.We extracted SNPs associated with the exposure that met the GWAS significance threshold (P < 5 × 10-8) to serve as instrumental variables (IVs). To ensure their independence, we extracted SNPs associated with the exposure that met the GWAS significance threshold (P < 5 × 10-8) to serve as instrumental variables (IVs), linkage disequilibrium (LD) clumping was performed using the ‘TwoSampleMR’ package (version 0.6.9) in R, with a strict LD threshold of r2 < 0.001 and a clumping window of 10,000 kb, extracted the corresponding data from the GWAS results (Table 1).

Table 1 Details on the characteristics of each included dataset.

Phenotype	Data source	Total sample size	Population	#SNPs	
C-reactive protein	Said et al. (2022)	575,531	European	11.1M	
low density lipoprotein	Sinnott-Armstrong et al. (2021)	355,197	European	9M	
Platelet count	Loh et al. (2018)	600,968	European	12M	
Spleen volume	Liu et al. (2021)	32,860	European	9.3M	
hepatocellular carcinoma	Jiang et al. (2021)	123 cases, 456,225 controls	European	11.8M	
Notes.

SNP, single nucleotide polymorphisms.

The GWAS summary data of HCC comes from UK Biobank (European ancestry, 123 cases, and 456,225 controls), containing a total of 11.8M SNPs. For detailed data such as genotyping, endpoint definitions, etc., please visit the UK Biobank webpage (https://www.ukbiobank.ac.uk/).

Nomogram

Our retrospective study collected the clinical data of 453 patients with cirrhosis/liver cancer who were treated at the Second Affiliated Hospital of Zhejiang University School of Medicine from October 2022 to October 2024, the final follow-up date for data collection was October 31, 2024. All patients were diagnosed according to the European Liver Association < <EASL Clinical Practice Guidelines for the management of patients with decompensated cirrhosis> > and < <EASL Clinical Practice Guidelines: Management of hepatocellular carcinoma> > (European Association for the Study of the Liver, 2018). All patients were diagnosed based on past medical history, clinical manifestations, laboratory examinations, imaging examinations or histopathological examinations, and all patients underwent abdominal CT or abdominal ultrasound examination. Inclusion criteria: older than 18 years old; patients diagnosed with hepatitis B cirrhosis; patients diagnosed with liver cancer caused by hepatitis B cirrhosis; with complete imaging or blood test data. Exclusion criteria: liver cirrhosis caused by other causes; combined with other tumor diseases; severe data loss; pregnancy or lactation; combined with severe cardiopulmonary and renal insufficiency. A total of 306 patients were excluded. A total of 147 patients were included in this study. File S2 presents the demographic and clinical characteristics of the study participants.

Clinical data and laboratory indicators: We collect patients’ gender, age, underlying diseases (hypertension, diabetes), antiviral treatment, body mass index (BMI), hepatitis B virus DNA quantification, LDL, abnormal prothrombin (PIVKA-II), alpha-fetoprotein (AFP), aspartate aminotransferase (AST), alanine aminotransferase (ALT), albumin and prothrombin time (PT). A-index obtained through AFP, ATL, ALT.Positive hepatitis B virus DNA quantification (“+”) is defined as HBV-DNA>1x10E3 copies/ml, and negative (“-”) is defined as HBV-DNA<1x10E3 copies/ml.All examinations are completed by the laboratory department of our hospital, and case data come from the hospital’s electronic medical record system. According to whether HCC occurs in patients with hepatitis B cirrhosis, the patients are divided into hepatitis B cirrhosis group and hepatocellular carcinoma group. Ethical approval for this study was obtained from the Ethics Committee of the Second Affiliated Hospital of Zhejiang University School of Medicine (2025-0275), with the research conducted in strict accordance with the principles of the Declaration of Helsinki.

Study design

Mendelian randomization

We employed a two-sample Mendelian randomization (MR) approach to investigate the potential relationships between clinical indicators and the incidence of hepatocellular carcinoma (HCC). MR studies require the following three assumptions to be satisfied: (1) Relevance assumption: Genetic variants must have a robust association with the exposure. (2) Independence assumption: Genetic variants should not be associated with confounding variables. (3) Exclusion restriction assumption: Genetic variants influence the outcome only through the exposure (Emdin, Khera & Kathiresan, 2017).

The study utilized the inverse-variance weighted (IVW) model to estimate causality, with additional validation using MR-Egger regression and the weighted median (WM) model. The IVW method assumes all SNPs are valid instrumental variables, providing more accurate results (Bowden & Holmes, 2019). MR-Egger regression detects pleiotropy but may be less precise due to genetic variation. The WM model is more robust to errors, offering stronger causal estimates under lower bias. A causal relationship was considered significant if the IVW analysis yielded a P-value < 0.05.

In this study, we employed MR-Egger regression for sensitivity analyses to detect horizontal pleiotropy within the genetic instruments (Bowden, Davey Smith & Burgess, 2015). Upon identification of pleiotropy, the implicated SNPs were removed, and the analyses were rerun. The heterogeneity among SNPs linked to HCC was appraised using Cochran’s Q test. A leave-one-out analysis was executed to determine the impact of individual SNPs on the overall estimate. The F statistic, calculated as F = R2 (N−k−1)/k(1−R2) (Burgess & Thompson, 2011), was used to evaluate instrument strength, where R2 denotes the genetic variants, N is the sample size, and k represents the number of instruments. Instruments with an F statistic <10 were excluded as weak instruments. All statistical analyses were performed using R software (version 4.3.3; R Core Team, 2024) and the TwoSampleMR package (version 0.6.9).

Restricted cubic strips and Principal component analysis

In many real-world datasets, the relationship between independent and dependent variables is often nonlinear. Traditional linear regression models are inadequate for capturing these nonlinear relationships; however, restricted cubic splines (RCS) can effectively model such complexities by fitting piecewise within the data range. We employed RCS to investigate the nonlinear relationship between LDL levels and HCC, 3 knots placed at the 10th, 50th, and 90th percentiles of the LDL distribution, using the rms package,allowing us to identify the turning point and further explore the association between LDL levels and HCC.

To reduce data dimensionality and mitigate multicollinearity among variables, we employed principal component analysis (PCA). By aggregating multiple related variables into a limited number of principal components, PCA effectively decreases data dimensionality, removes noise, and enhances data quality. In this study, we standardized the data prior to applying PCA to extract the principal components. PCA was carried out using the FactoMineR package. A varimax rotation was applied to maximize variance interpretation, and the first principal component was retained as a composite index. Ultimately, we will utilize these principal components for subsequent regression analysis.

Logistic regression and nomogram

Continuous variables were expressed as mean ± standard deviation (x ± s) and analyzed using t-tests for intergroup differences. Categorical variables were expressed as percentages and analyzed using chi-square tests. To identify risk factors for HCC development in patients with HBV-related cirrhosis, univariate and multivariate logistic regression analyses were performed.

Receiver operating characteristic (ROC) curves were plotted to calculate the area under the curve (AUC), and the optimal cutoff value was determined based on sensitivity and specificity. A predictive nomogram for HCC occurrence was constructed based on the regression model. Model performance was evaluated using calibration plots and decision curve analysis. Internal validation was performed using the bootstrap method with 1,000 repetitions. Statistical significance was defined as P < 0.05.

Results

Mendelian randomization analysis of the causal relationship between exposure factors and hepatocellular carcinoma

(1) Characteristics of genetic instrumental variables

In the CRP population, 256 SNPs related to HCC were screened out, and in the LDL population, 205 SNPs were screened out. The total number of platelets 595 SNPs were screened, and 31 SNPs were screened for spleen volume.

(2) Correlation with HCC

The causal relationship between LDL and HCC is shown in Table 2. The random effects IVW results showed that the reduction of LDL was associated with an increased risk of HCC (ORIVW = 0.472, 95% CI [0.259–0.860]; P = 0.014), and showed heterogeneity by Cochran Q test (Q = 260.6, P = 0.0013). MR-Egger regression found no evidence of directional pleiotropy (intercept p-value = 0.095) (Table 2).

Table 2 Causal effects of clinical indicators on HCC.

LDL	IVW method	MR Egger	Weighted median	Cochran’s Q	
OR (95% CI)	P-value	OR (95% CI)	P-value	OR (95% CI)	P-value	P-value	
0.472 (0.259–0.860)	0.014	0.49 (0.213–1.1276)	0.095	0.301 (0.127–0.713)	0.006	0.001371559	
CRP	IVW method	MR Egger	Weighted median		
OR (95% CI)	P-value	OR (95% CI)	P-value	OR (95% CI)	P-value	P-value	
1.146 (0.519–2.532)	0.734895	0.581 (0.143–2.358)	0.449	0.533 (0.141–2.016)	0.36	0.4129912	
PC	IVW method	MR Egger	Weighted median		
OR (95% CI)	P-value	OR (95% CI)	P-value	OR (95% CI)	P-value	P-value	
1.257 (0.789–2.002)	0.335	0.872 (0.393–1.933)	0.737	1.149 (0.531–2.487)	0.723	0.5772943	
SV	IVW method	MR Egger	Weighted median		
OR (95% CI)	P-value	OR (95% CI)	P-value	OR (95% CI)	P-value	P-value	
0.754 (0.229–2.484)	0.643	0.666 (0.037–11.793)	0.784	0.669 (0.146–3.054)	0.592	0.06817777	
Notes.

LDL, low density lipoprotein; CRP, C-reactive protein; PC, Platelet count; SV, Spleen volume.

(3) Leave-one-out analysis

Leave one out analysis results show that after gradually eliminating each SNP, the causal relationship of LDL to HCC is not affected by individual SNPs and still shows significance, which shows that the results are robust and reliable (Fig. 2).

Figure 2 Mendelian randomization.

(A) Research design overview. (B) Funnel plot of LDL and HCC. (C) The main impact of LDL on HCC. (D) Leave-one-out.

For CRP, PC and PV, it has not yet been determined whether they have an inverse correlation with HCC, and further research is needed. See Files S1–S4 for more information.

Non-linear analysis of LDL and HCC and new forecasting factor developed through PCA

Our findings indicate a nonlinear relationship between LDL and HCC (LDL: P = 0.0003; Nonlinear: P = 0.0004), with a turning point identified at 2.28 mmol/L. Below this threshold, LDL and HCC are positively correlated; conversely, when LDL levels are equal to or exceed 2.28 mmol/L, the risk of HCC decreases as LDL levels increase. We synthesized AFP, ATL, and AST into a new predictive factor termed the ‘A index’ through PCA, assigning each case its unique value, which was subsequently analyzed using both univariate and multivariate logistic regression.

Risk factors and nomogram for hepatocellular carcinoma in patients with hepatitis B cirrhosis

Among all 147 patients with hepatitis B cirrhosis, 102 developed hepatocellular carcinoma. There were differences in LDL, antiviral treatment, gender, age, BMI, AFP, albumin, aspartate aminotransferase and PT between the two groups of patients, and other characteristics were similar (P > 0.05) (Table 3). We used univariate logistic regression to explore 14 factors in multivariate logistic regression (abnormal prothrombin in patients with hepatitis B cirrhosis has insufficient data, so it was not included in the analysis), using a stepwise backward method, age (odds ratio (OR): 1.053, 95% confidence interval (CI) [1.004–1.105], P = 0.031), sex (7.464; 2.010, 27.708; P = 0.002), PT (0.385; 0.239, 0.621; P < 0.001), A-index (4.633; 1.579, 13.595; P = 0.005), antiviral treatment (0.236; 0.073, 0.756; P = 0.015) was proven to be an independent predictor of hepatocellular carcinoma in patients with hepatitis B cirrhosis. Since Mendelian randomization proved that LDL and HCC have a causal relationship, LDL will also be included in the prediction model (Table 4).

Table 3 Baseline table.

	Hepatitis B-related cirrhosis	Hepatocellular carcinoma	p	
n	45	102		
LDL (mean (SD))	2.04 (0.85)	2.35 (0.58)	0.012	
Antiviral_treatment(%)	33 (73.3)	43 (42.2)	0.001	
Sex(%)			0.028	
Male	33 (73.3)	91 (89.2)		
Female	12 (26.7)	11 (10.8)		
Age (mean (SD))	55.18 (12.69)	59.77 (10.75)	0.025	
Abnormal_prothrombin (mean (SD))	N/A	6584.95 (19905.74)	N/A	
BMI (mean (SD))	24.13 (3.86)	21.98 (3.17)	0.001	
Hepatitis_B_virus_DNA_detection (“+”)(%)	5 (11.1)	9 (8.8)	0.896	
AFP (mean (SD))	9.48 (35.88)	1649.06 (4371.39)	0.013	
Albumin (mean (SD))	33.00 (5.39)	36.51 (5.64)	0.001	
AST (mean (SD))	42.16 (19.77)	74.90 (97.01)	0.027	
ALT (mean (SD))	30.47 (20.90)	49.25 (67.01)	0.068	
PT (mean (SD))	15.92 (1.82)	13.89 (1.34)	<0.001	
Diabetes (%)	4 (8.9)	11 (10.8)	0.957	
Hypertension (%)	7 (15.6)	30 (29.4)	0.115	

Table 4 Univariate and multivariate logistic regression.

Variables	Univariate	Multivariable	
	OR	95% CL	P	OR	95% CL	P	
LDL	1.98	1.150–3.410	0.013				
Antiviral_treatment	0.265	0.122–0.571	0.0007	0.236	0.073–0.756	0.015	
Sex	3.008	1.210–7.472	0.017	7.464	2.010–27.708	0.002	
Age	1.036	1.003–1.069	0.027	1.053	1.004–1.105	0.031	
BMI	0.838	0.754–0.932	0.001				
AFP	1.021	1.004–1.038	0.011				
Albumin	1.119	1.046–1.196	0.001				
AST	1.017	1.003–1.031	0.012				
ALT	1.016	1.001–1.031	0.035				
PT	0.452	0.338–0.605	<0.001	0.385	0.239–0.621	<0.001	
A-index	3.039	1.408–6.559	0.004	4.633	1.579–13.595	0.005	
Giabetes	1.239	0.372–4.123	0.727				
Hypentension	2.262	0.909–5.629	0.079				
HBVDD	0.774	0.244–2.456	0.664				

A post hoc power analysis was performed using the pwr package (version 1.3-0) in R. Given the total sample size (n = 147) and the large observed effect sizes for key predictors in the final multivariable model (e.g., OR of 7.464 for sex and 0.385 for PT), the analysis confirmed that the study had excellent statistical power (>99%) to detect these effects at a significance level (α) of 0.05.

Construction and validation of prediction model

Based on the above factors, we constructed a nomogram to predict the occurrence of hepatocellular carcinoma in patients with hepatitis B cirrhosis. The scoring scale represents the score of each factor. The sum of the scores of the five factors will give a total score, and the corresponding probability is the probability of hepatocellular carcinoma in patients with hepatitis B cirrhosis (Fig. 3).

We validated the nomogram. The AUC value of each factor is as follows: antiviral treatment (AUC = 0.655), age (AUC = 0.610), PT (AUC = 0.838), Sex (AUC = 0.579), LDL (AUC = 0.651), A-index (AUC = 0.652), the comprehensive model has the highest AUC (AUC = 0.938), the optimal cutoff value was 0.745. The model was calibrated and a calibration curve was drawn. The bootstrap method was repeated 1,000 times. The close proximity between the actual prediction ability line and the ideal line indicates that the model has good consistency. Decision curve analysis (DCA) demonstrate the net benefit of this model in a clinical setting, indicating excellent clinical results (Fig. 4).

Figure 3 Nomogram about HCC.

Figure 4 The reliability of the nomogram.

(A) RCS curve of LDL. (B) Nomogramprediction model was evaluated by the ROC curve. (C) Decision curve analysis of nomogram predictive models. (D) Calibration curve of the nomogram prediction model.

Discussion

The liver functions as the body’s “biochemical factory”. The occurrence of liver cancer leads to the disruption of normal biochemical processes. As a result, metabolic irregularities along with inflammatory responses may influence the progression of liver cancer. Various studies indicate that C-reactive protein (CRP) could act as a biomarker for the diagnosis of hepatitis B virus (HBV)-related HCC in patients who test negative for alpha-fetoprotein (AFP) (She et al., 2015). Platelets have the potential to create a supportive microenvironment for tumor proliferation. In mouse models of HCC, activated platelets attach to tumor cells and, through their structural network, conceal the tumor cells, enabling them to avoid immune detection and metastasize effectively (Coupland, Chong & Parish, 2012; Zhuang et al., 2019). Low-density lipoprotein (LDL), which functions in cholesterol transport, is thought to be linked to HCC development. Abnormalities in lipid metabolism and associated enzymes may offer opportunities for the early diagnosis of HCC (Buechler & Aslanidis, 2020). Beyond biomarkers, the volume of the spleen, a crucial parameter indicating liver health and portal pressure, is associated with cirrhosis and has been identified by Dai et al. (2021) as a vital prognostic indicator for HCC (Son et al., 2020). Through the method of Mendelian randomization, we explored the connections between these factors and HCC, revealing that individuals exhibiting lower LDL levels had a notably heightened risk of HCC. This finding eliminates the common confounding variables found in typical observational studies and was corroborated through multivariable regression analysis. Consequently, we integrated traditional risk factors (including age, gender, prothrombin time, antiviral treatment and A-index) to develop a comprehensive multifactorial nomogram model. This approach represents an innovative effort to merge genetic and clinical data, aiming to create a risk assessment tool for HCC.

LDL is a type of lipoprotein that is produced and released by the liver, serving the role of transporting lipids, and it exhibits a complex relationship with HCC. According to Chen et al. (2021), a decrease in LDL receptors may promote intracellular cholesterol production through the MEK/ERK signaling pathway, thus facilitating the growth and movement of liver cancer cells. This phenomenon could be linked to the provision of essential biosynthetic materials for tumor cells by LDL and its associated lipids (Zhang et al., 2024). However, contrasting findings have emerged from other research. A retrospective analysis in a Korean cohort indicated that high cholesterol levels were associated with a reduced occurrence of HCC (Sinn et al., 2020), which aligns with results from another prospective investigation in China. This latter study revealed that when LDL cholesterol levels fell below 100 mg/dl, the risk of developing cancer increased (1.20 (1.08–1.34)) (Li et al., 2020). One plausible explanation for this is that liver cancer cells, when failing to undergo apoptosis, may continuously proliferate (Guicciardi et al., 2013). Elevated cholesterol along with low ceramide concentrations may stimulate cell proliferation and offer protection against oxidative stress, which in turn can diminish apoptosis (Buechler & Aslanidis, 2020). These findings contradict earlier research, prompting us to consider that LDL and HCC may exhibit non-linear relationships. Utilizing the RCS method, we identified a significant turning point for LDL. This observation aligns with the results we randomly noted through Mendelian analysis in the European population, which indicated that lower LDL levels increase the risk of HCC. Specifically, when LDL levels are ≥ 2.28 mmol/L, the probability of developing HCC decreases. This suggests that employing LDL as a variable in predictive models is effective for individuals in both East Asia and Europe. Interestingly, a recent large cohort study linked statin (HMG-CoA reductase inhibitors) use to a reduced risk of HCC—a finding that appears to contradict our genetic results (Choi et al., 2025). This discrepancy may be explained by the pleiotropic effects of statins, which extend beyond LDL reduction to include anti-inflammatory, anti-angiogenic, and pro-apoptotic mechanisms that may independently suppress hepatocarcinogenesis. Thus, while genetic predisposition to low LDL may increase risk, pharmacological modulation via statins likely confers net protection through multiple pathways. These insights reinforce the clinical utility of our nomogram in identifying high-risk patients who may benefit from intensified surveillance, including potential statin therapy.

HCC presents distinct epidemiological patterns influenced by age and gender. Incidence remains low before age 40, progressively increasing to peak at 70–75 years (El-Serag, 2004). Notably, younger populations demonstrate heightened HBV susceptibility due to immature immunity, leading to delayed tumor detection and advanced-stage diagnosis (Tandon & Garcia-Tsao, 2009; Bosetti, Turati & La Vecchia, 2014). Paradoxically, patients aged <65 exhibit superior 5-10 year survival compared to elderly counterparts (≥65 years) with comorbidities, potentially reflecting age-related immunosenescence (Su et al., 2012; Yoo et al., 2023). Gender disparities are equally striking: global HCC incidence shows 3–5:1 male predominance, with males demonstrating poorer advanced-stage prognosis (Liou et al., 2023). This dimorphism may stem from sex hormone interactions. Estrogen appears protective through PTPRO-mediated AKT/mTOR/SREBP1/ACC1 pathway regulation, evidenced by improved survival in oral contraceptive users and elevated PTPRO expression in female peri-tumoral tissues (Dai et al., 2022; Hou et al., 2013). Conversely, androgens promote hepatocarcinogenesis via multiple mechanisms: androgen receptor (AR) overexpression stimulates malignant hepatocyte proliferation through CCRK/β-catenin/TCF signaling in murine models (Ma et al., 2008; Feng et al., 2011), while AR-HBV enhancer interactions establish a carcinogenic feedback loop through sustained viral replication (Wang, Chen & Yeh, 2015; Wang et al., 2009). Multivariate analysis confirms age and gender as independent predictors in HBV-related cirrhotic patients, with HCC risk escalating with age and male gender. Integration of these parameters enhances nomogram predictive capacity for population-level applications.

The coagulation cascade exhibits bidirectional interplay with hepatocarcinogenesis and progression. Hepatic-derived coagulation factor VII (FVII) mediates tumor-stroma interactions via PAR2-dependent signaling, promoting tumor proliferation and metastasis (Chen et al., 2016; He, Yang & Jin, 2022). Prothrombin time (PT), a critical coagulation parameter, reflects both liver dysfunction and tumor progression. Competing risk analysis in cirrhotic cohorts revealed prolonged PT inversely correlates with HCC risk (Wang et al., 2023), consistent with our multivariable regression results. Paradoxically, Wang et al. (2017) demonstrated that PT ≥ 12.1 s predicts reduced survival, suggesting prognostic duality influenced by confounders like hepatic functional reserve. Given PT’s dual role in liver pathophysiology, we integrated it into our predictive model to systematically assess its impact on hepatocarcinogenesis and outcomes.

Chronic HBV infection subverts immune surveillance through NK cell cytotoxic impairment (Zheng, Dou & Wang, 2023). Viral genome integration sustains HBx-driven oncogenic transcription and hepatocyte proliferation (Zamor, deLemos & Russo, 2017), while persistent infection induces hepatocyte injury-regeneration cycles, fostering mutagenesis through inflammatory microenvironments (Zamor, deLemos & Russo, 2017). Nucleos(t)ide analogs (NAs; e.g., entecavir, tenofovir) suppress HBV replication (6-log DNA reduction/year) (Liang et al., 2015), restoring hepatic immunosurveillance to reduce HCC risk. Select NAs like 2-CdA exhibit direct antitumor effects via HepG2 proliferation inhibition (Graziadei et al., 1998). Antiviral regimens additionally attenuate fibrogenesis and improve hepatic microenvironmental homeostasis (Bruni et al., 2024). Multivariate analysis confirmed anti-HBV therapy independently inversely correlates with HCC incidence. Our predictive model integrates these pharmacological mechanisms with demographic determinants (age/gender) for personalized risk stratification and therapeutic alignment.

AFP is a protein synthesized in the liver, primarily during fetal development. In adults, AFP levels are typically very low; however, they may become abnormally elevated in certain diseases. An AFP level slightly higher than the normal range (6–19 ng/ml) may indicate an increased risk for HCC (Taura et al., 2012). Furthermore, incorporating the detection of AST and ALT into the screening process for AFP can significantly enhance the detection capacity for liver cancer (Tayob et al., 2021). Although some studies have suggested the diagnostic performance of the “ratio of AFP to transaminase” for HCC, the involvement of multiple variables introduces the potential for subjective weighting by human judgment (Liu et al., 2019). In contrast, the PCA method we employ is grounded in statistical principles, allowing for the objective extraction of primary variance information from the data while simultaneously reducing complexity and noise. The “A Index”, which integrates AFP, AST, and ALT, more effectively captures the inherent relationships and overall trends among these variables, thereby enhancing the accuracy of predictive models.

This research investigated the causal pathways and risk prediction frameworks for HCC in patients with HBV-related cirrhosis, employing Mendelian randomization alongside retrospective clinical datasets. Nevertheless, several limitations should be considered in this study. We note that the HCC case count in the outcome GWAS may not be sufficient, resulting in a low instrument strength (mean F-statistic = 1.33) and possibly inflating the uncertainty of our MR estimate. Although hinting at causality, the result requires careful interpretation and future replication. The relatively small cohort size and the absence of certain variables—such as abnormal prothrombin—due to limited data availability may affect the generalizability of our findings. Although the model incorporated genetic and clinical biomarkers, it did not include environmental variables like lifestyle or dietary habits. Moreover, recent studies have emphasized the prognostic value of imaging-based vascular parameters (e.g., portal venous coefficients) for predicting postoperative survival (Li et al., 2024b), as well as features of the local tumor immune microenvironment such as lymphatic vessel density, which has emerged as a relevant biomarker in HCC (Li et al., 2024a); however, these were not integrated into our current model. While the nomogram demonstrated strong performance in internal validation, external validation remains essential. Future prospective studies, ideally with a multi-center design, should be conducted to validate and refine our model. Further efforts should also aim to combine readily available clinical-genetic predictors with radiological and histopathological data within a prospective framework to develop a more integrated and powerful tool for pre- and postoperative risk stratification. Larger prospective cohorts and the systematic incorporation of multimodal data—including environmental, radiological, and histopathological metrics—will be crucial to enhance the predictive accuracy and clinical applicability of the model in HCC.

Conclusion

This study demonstrated a causal association between LDL levels and HCC risk in patients with hepatitis B virus (HBV)-related cirrhosis. Through PCA, we derived a novel predictive metric, the A-index, which facilitated the development of a robust prognostic model integrating conventional risk factors. The synergistic incorporation of genetic and clinical biomarkers provides a transformative risk-stratification tool, directly informing clinical practice by underscoring the need for nuanced metabolic management and intensified surveillance for high-risk individuals. Nevertheless, the clinical adoption of this model faces barriers such as integrating genetic profiling into routine workflows. Prior to implementation, external validation in diverse multicenter cohorts remains imperative to confirm its generalizability and utility.

Supplemental Information

Supplemental Information 1 Raw data from clinical studies

Supplemental Information 2 Data on Mendelian randomization

Supplemental Information 3 STROBE checklist

Abbreviation

HBV-C Hepatitis B viral cirrhosis

HCC hepatocellular carcinoma

SNP Single Nucleotide Polymorphism

MR Mendelian randomization

HBV Hepatitis B virus

PCA principal components analysis

RCS restricted cubic spline

ROC Receiver operating characteristic

AUC Area Under Curve

IV instrumental variable

IVW inverse-varianceweighted

OR odds ratio

CI confidence interval

GWASs Genome-wide association studies

AFP alpha-fetoprotein.

CRP C-reactive protein

LDL low density lipoprotein

OX–LDL Oxidized Low-density Lipoprotein

SV spleen volume

PIVKA-II abnormal prothrombin

BMI body mass index

AST aspartate aminotransferase

ALT alanine aminotransferase

PT prothrombin time

Alb albumin

WM Weighted Median

DCA Decision Curve Analysis

PTPRO Protein Tyrosine Phosphatase Receptor Type O

PTP protein tyrosine phosphatase

AR androgen receptor

NK cell natural killer cell

HBx Hepatitis B virus X protein

NAs Nucleos(t)ide analogs

HBVDD Hepatitis B virus DNA detection

Additional Information and Declarations

Competing Interests

Author Contributions

Human Ethics

Data Availability

The authors declare there are no competing interests.

Xiaolong Zheng conceived and designed the experiments, performed the experiments, analyzed the data, prepared figures and/or tables, and approved the final draft.

Yiping Hong conceived and designed the experiments, performed the experiments, prepared figures and/or tables, and approved the final draft.

Wei Wei conceived and designed the experiments, authored or reviewed drafts of the article, and approved the final draft.

The following information was supplied relating to ethical approvals (i.e., approving body and any reference numbers):

All patients followed the principles of the Declaration of Helsinki, and the study protocol was approved by the Ethics Committee of the Second Affiliated Hospital of Zhejiang University School of Medicine (2025-0275). The requirement for informed consent for the use of clinical records was waived due to the retrospective study design. However, all patients were guaranteed opportunities to learn about the study and withdraw from it through notification.

The following information was supplied regarding data availability:

The data is available in the Supplemental Files.

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
