# Peer review of "Mendelian randomization and nomogram-based prediction of hepatocellular carcinoma risk in patients with hepatitis B cirrhosis"

_PeerJ, doi:10.7717/peerj.20179_

## Round 0.1 · original submission · Minor Revisions

I'm impressed with your manuscript. Please consider the papers mentioned by Reviewer 1 and whether they might lead you to add anything to the MS. Reviewer 2 has a number of suggestions to improve the MS; I believe all of them can be incorporated relatively easily.

·

Basic reporting

The study integrates genetic and clinical data to predict HCC risk in HBV-C patients. Using Mendelian Randomization and a retrospective cohort (n=147), it identifies a critical LDL threshold at 2.28 mmol/L and develops an "A-index" via PCA. The resulting nomogram shows high accuracy (AUC=0.938) and clinical utility, offering a novel tool for personalized HCC risk stratification.

1. The reviewer suggests that authors read "Journal of Hepatocellular Carcinoma 2024, 11: 1389-1402" for the importance of vascular factors, such as portal venous and hepatic arterial coefficients, in predicting postoperative survival in HCC patients.

2. The reviewer suggests that reading "Frontier in Immunology. 2025; 15: 1519999" is relevant for the potential of tumor-associated lymphatic vessel density as a postoperative prognostic biomarker in hepatobiliary cancers.

Experimental design

The study aims to investigate the causal relationships between metabolic-inflammatory biomarkers and HCC risk using Mendelian randomization (MR) and to develop a precision prediction model integrating genetic evidence with nonlinear biochemical dynamics.

The authors utilized a two-sample MR approach on GWAS datasets from the UK Biobank and a retrospective HBV-C cohort.

The rationale for using MR is sound, as it helps mitigate confounding factors often present in observational studies.

The use of restricted cubic splines(RCS)and principal component analysis (PCA) to identify nonlinear relationships and reduce data dimensionality is innovative and appropriate for the study's objectives.

Validity of the findings

The findings reveal a robust causal link between reduced low-density lipoprotein(LDL)levels and elevated HCC risk, with a critical LDL threshold identified at 2.28 mmol/L.

The PCA-derived "A-index" outperformed individual biomarkers in predicting HCC risk.

The final nomogram integrating LDL dynamics, A-index, and clinical predictors achieved high discrimination(AUC=0.938)and clinical net benefit.

These results are statistically sound and well-supported by the data.

Additional comments

The study focuses primarily on genetic and clinical indicators, with limited consideration of environmental factors and their interactions with HCC. Future research should incorporate multi-omics approaches.

Reviewer 2 ·

Basic reporting

Title & Abstract
• The title has effectively captured the study’s use of both MR and clinical prediction models for stratifying HCC risk in patients with HBV-related cirrhosis. However, the phrase “from genes to clinical indicators” is complex; hence, rewrite.
• The abstract has provided a concise summary of the study’s background, methods, main findings, and conclusion.
• To improve the clarity, briefly define the A-index. Additionally, include details about the cohort.

Introduction
• The introduction has clearly outlined the prevalence and significance of HCC in patients with HBV-related cirrhosis. However, it can be strengthened by referencing recent MR research (eg, https://doi.org/10.1186/s12944-023-01877-1).
• While the authors have explained the motivation for integrating MR with the PCA-nomogram approach, it would benefit from a discussion of the shortcomings of existing HCC risk nomograms. This will help clarify the value of the study’s methodology.

Figures & Tables
• The figures are easy to interpret, but please make sure that all axis labels, such as those on spline curves, clearly include units for better understanding.
• There’s no evidence of data manipulation. To support transparency, please ensure that raw data, such as the SNP effects, are included in Supplementary tables S1-S4.
• Some tables in the PDF are quite small; increasing the font size or splitting the large tables will help improve readability.

Experimental design

Material and Methods
• GWAS sources are identified, but the selection criteria for SNP and LD-clumping parameters should be cited with specific software and version.
• Please provide a knot placement strategy for RCS. Also specify the PCA package and rotation method.
• Report the demographic and clinical summary, as well as the censoring dates.
• Choice of retrospective cohort size might limit the detection. Power calculation for logistic regression should be included.
• The MR sample is adequate for primary analysis; however, the HCC cases are relatively few, which potentially affects the SNP-HCC F-statistics.

Validity of the findings

Results
• The study’s causal MR finding that lower LDL cholesterol increases HCC risk is aligned with previous research. This study is among the first that directly incorporates this genetic causality into a clinical risk nomogram.
• Sensitivity analyses, including the MR-Egger regression and leave-one-out tests, are well executed and support the findings. It can be strengthened if authors present full MR-Egger Intercept P-values in the main text.
• A combination of MR-derived genetic causality with A-index helps in achieving stronger predictive discrimination than traditional nomograms.

Discussion
• The discussion clearly explains the non-linear relationship between LDL and HCC risk, hence helps comparing both European and Asian mendelian randomization studies.
• The authors did a good job connecting factors like sex, PT time, age, and anti-viral therapy to the immunological and hormonal pathways relevant to HCC development.
• To further enrich the context, the authors can mention the recent research suggesting that statins may lower HCC risk, which will help in framing the clinical implications of the study.
• Authors should acknowledge the limitations of the study, such as the single-center nature and potential selection bias.

Conclusion
• The study integrates the genetic causality and clinical variables using MR and PCA to construct the HCC risk nomogram for HBV-cirrhosis patients. A few recommendations:
• Emphasize how the MR findings directly apprise the modifiable risk factors and guide both the pharma and non-pharma interventions in clinical practice.
• Clearly mention the need for external validation in diverse, multicenter cohorts.
• Mention potential barriers to clinical adoption, such as the incorporation of genetic risk profiling into routine hepatology workflows.

---

## Round 0.2 · accepted · Accept

I have reviewed the revisions and I'm happy with the current version.